# Mussel larvae modify calcifying fluid carbonate chemistry to promote calcification

Kirti Ramesh[1], Marian Y. Hu[2], Jörn Thomsen[1], Markus Bleich[2] & Frank Melzner[1]

Understanding mollusk calcification sensitivity to ocean acidification (OA) requires a better knowledge of calcification mechanisms. Especially in rapidly calcifying larval stages, mechanisms of shell formation are largely unexplored—yet these are the most vulnerable life stages. Here we find rapid generation of crystalline shell material in mussel larvae. We find no evidence for intracellular $CaCO_3$ formation, indicating that mineral formation could be constrained to the calcifying space beneath the shell. Using microelectrodes we show that larvae can increase pH and $[CO_3^{2-}]$ beneath the growing shell, leading to a ~1.5-fold elevation in calcium carbonate saturation state ($\Omega_{arag}$). Larvae exposed to OA exhibit a drop in pH, $[CO_3^{2-}]$ and $\Omega_{arag}$ at the site of calcification, which correlates with decreased shell growth, and, eventually, shell dissolution. Our findings help explain why bivalve larvae can form shells under moderate acidification scenarios and provide a direct link between ocean carbonate chemistry and larval calcification rate.

[1] GEOMAR Helmholtz Centre for Ocean Research Kiel, 24148 Kiel, Germany. [2] Institute of Physiology, Christian-Albrechts-University Kiel, 24098 Kiel, Germany. Correspondence and requests for materials should be addressed to K.R. (email: kramesh@geomar.de) or to F.M. (email: fmelzner@geomar.de)

Understanding the impacts of ocean acidification (OA) on reef forming bivalves (oysters, mussels) is an important challenge, as these foundation species provide several essential ecosystem services[1]. Adult life stages of several bivalve species have been shown to be relatively vulnerable to OA[2–4]. However, bivalve larvae are particularly sensitive and respond negatively through impaired growth and calcification[5], increased shell malformation[6] and dissolution[7], as well as increased mortality[3]. Calcification performance of larval stages might be the main bottleneck determining vulnerability of bivalves to ongoing OA. However, despite existing information on the impacts of OA on bivalves, the physiological mechanisms driving their sensitivity to OA remain unknown.

In marine systems, calcification is highly dependent on external seawater carbonate chemistry[8, 9]. Many marine species are primarily impacted by OA induced changes in body fluid $pCO_2$ and associated secondary acid–base regulatory processes that can impair fitness relevant traits, including development and calcification[10, 11]. However, shell formation rate of early bivalve larvae with very high relative calcification rates correlates best with the seawater saturation state of aragonite ($\Omega_{arag}$)[9] or equivalently, the ratio of bicarbonate to protons ($[HCO_3^-]/[H^+]$)[8]. This is due to

**Fig. 1** Calcium accumulation and in vivo larval shell formation from Exp 1 and 2. **a** larval calcium accumulation during the first 48 hpf, N = 4 fertilizations, means and s.d. Shells are birefringent from 21 hpf onwards. **b** Confocal projection of early stage of shell formation in 21 hpf trochophore larva, calcein positive particles are deposited onto tissue facing side of organic shell cover and numerous calcein positive intracellular vesicles are visible underneath. Shell organic cover boundaries indicated by dashed line. **c** Same larva, merged transmission and calcein fluorescence images. **d** confocal projection of 28 hpf larva viewed from the hinge, which is not calcified yet. Shell fluorescence overexposed to visualize intracellular calcein positive vesicles. **e** Same larva imaged at lower intensity to illustrate shell details, calcein positive particles visible in the centre of each valve, dotted line indicates boundaries of the organic shell cover. **f** Close-up of calcein positive deposits attached to the organic shell cover in 21 hpf larva, organic cover boundary indicated by dotted line. **g** Same larva, transmission image. **h** Confocal projection of shell material secreted by 24 hpf larva, note deposits at the centre of the valve (c) and at the growing edge (ge). **i** Larvae in vivo under crossed polarized light, see also Supplementary Movie 1 for swimming and rotating larvae. **j** Same larvae, transmitted light. Scale bars: 20 μm **b**–**e**, 5 μm **f**–**h**, 75 μm **i**, **j**

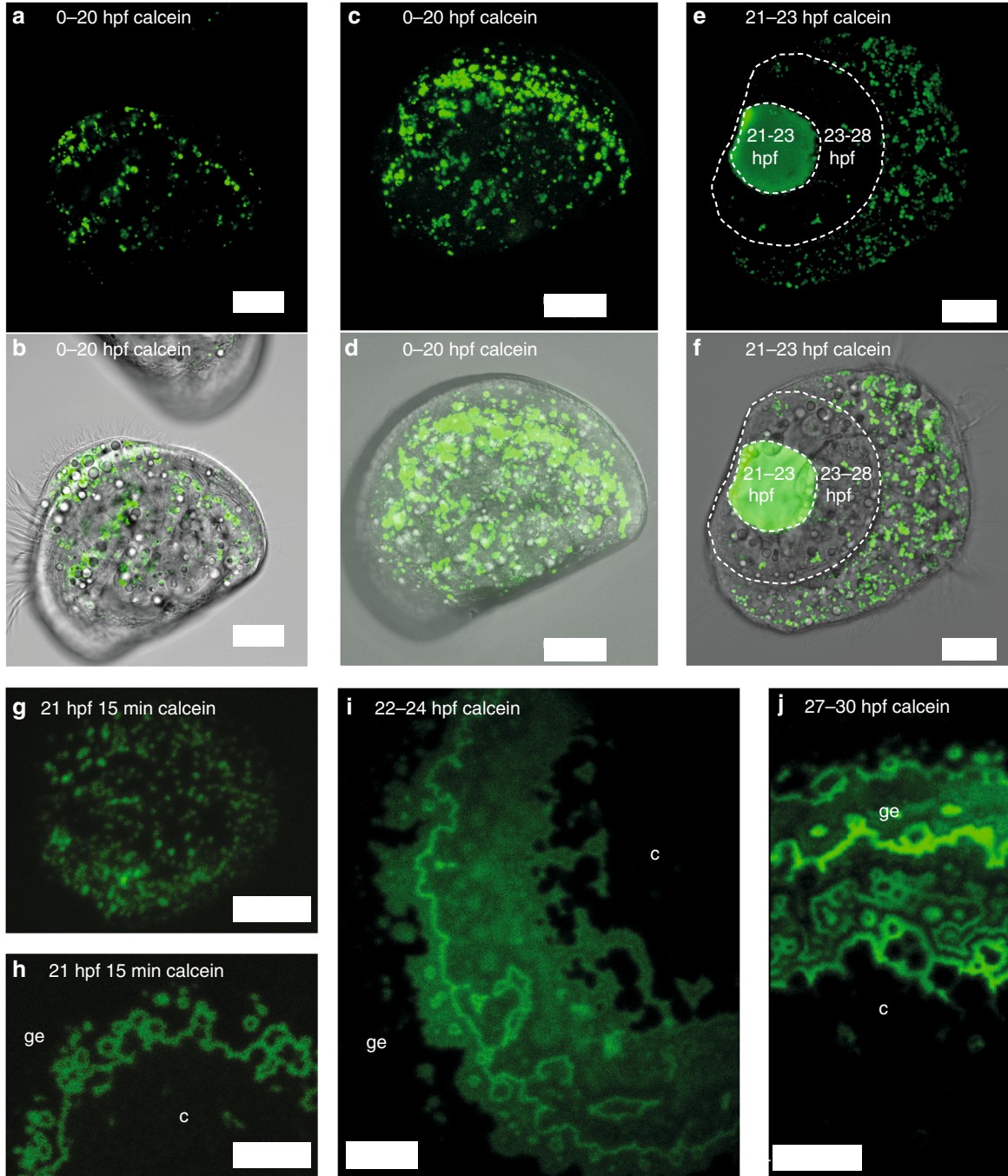

**Fig. 2** Calcein pulse-chase experiments and in vivo confocal microscopy from Exp 2. **a**, **b** Calcein fluorescence image and merged fluorescence and transmission images of larva cultured in calcein between 0–20 hpf, followed by development in filtered seawater (FSW) until 48 hpf, 1 μm thick section through body and shell to illustrate lack of calcein fluorescence in the shell. **c**, **d** Same animal, confocal projection through one entire shell valve and body to illustrate numerous calcein positive intracellular vesicles yet no fluorescence in the shell. **e**, **f** Confocal projection of calcein fluorescence of larva cultured in calcein FSW between 21–23 hpf. Shell material formed between 21–23 hpf is calcein labeled, shell material formed between 23–28 hpf is not. Calcein label of vesicles present at 28 hpf has apparently not been transferred into the shell. **g**, **h** 21 hpf animals stained with calcein for 15 min, then washed and cultured in FSW, calcein positive particles on periostracum **g**, shell growth bands in a slightly more advanced larva from the same fertilization **h**. **i** Confocal projection of shell calcein fluorescence of larva cultured in seawater with calcein pulse between 22–24 hpf. **j** Confocal projection of shell calcein fluorescence of larva cultured in seawater with calcein pulse between 27–30 hpf. c = centre of shell valve, ge = growing edge of shell valve. Scale bars: 20 μm **a–f**, 5 μm **h**, **i**, 10 μm **j**

the need to accumulate high amounts of calcification substrates ($HCO_3^-$, $Ca^{2+}$) and to excrete equivalent amounts of end products ($H^+$). Early larval stages (trochophore larvae) form a shell field during the first day of life, which then grows over the larval body within the next day until the first larval shell, the so-called prodissoconch I (PD I) is completed (veliger stage). During this brief time interval, bivalve larvae can precipitate a shell mass that is almost equivalent to their own body mass[12].

It is unknown at present whether the substrates for calcification ($HCO_3^-$, $Ca^{2+}$) are directly transferred via transcellular or paracellular routes to the calcification space (CS), the extracellular space directly underneath the growing shell. Alternatively, it is

also possible that calcification substrates in larvae are accumulated intracellularly in vesicles, which are then deposited into the growing shell via exocytosis. Presence of intracellular crystalline calcite and deposition onto the shell has been observed in oyster hemocytes during shell repair[13, 14]. Intracellular formation of amorphous calcium carbonate (ACC) precursors could be demonstrated in sea urchin larvae[15], where numerous large vesicles >1 μm accumulate calcium carbonate for subsequent deposition into the larval skeleton, where it then transforms to calcite[16]. In bivalve larvae, no such observations of intracellular ACC formation have been attempted. However, there is circumstantial evidence for a minor ACC fraction in larval bivalve shells that might be a precursor to aragonite, the major $CaCO_3$ polymorph of the shell[17]. This would support the intracellular ACC formation hypothesis. Intracellular ACC formation could be beneficial during OA stress, as intracellular pH typically is much more tightly regulated than extracellular pH, thereby providing stable conditions for vesicular carbonate precipitation[18].

It is also unclear, to what extent bivalve larvae can control the carbonate chemistry of the extracellular CS underneath the shell. OA induced changes in carbonate chemistry at the CS could lead to internal shell dissolution and reduced rates of shell deposition. The chemical composition of this space should govern precipitation and dissolution rates (r) for inorganic carbonates according to:

$$\mathbf{r} = k(\Omega_{arag} - 1) \qquad (1)$$

where 'k' is the rate constant[9, 19]. For a given system with $k =$ constant, $\Omega_{arag}$ determines calcification rates and $r = 0$ when $\Omega_{arag} = 1$. However, dissolution of biogenic aragonite in bivalve shells has been empirically observed to occur at $\Omega_{arag} > 1$[20], likely due to the higher solubility of biogenic carbonates. On the other hand, bivalve larvae have been observed to develop shells even when the surrounding seawater is undersaturated with respect to $\Omega_{arag}$[6, 9]. Therefore, it has been suggested that larvae are capable of elevating $\Omega_{arag}$ at the site of calcification via active ion transport mechanisms[9]. Alternatively, modulation of the fraction and type of organic shell matrix molecules might enable continued calcification when $\Omega_{arag} < 1$[9]. In order to obtain a mechanistic understanding of the processes leading to tolerance vs. vulnerability to OA, it is paramount to obtain direct estimates of the carbonate chemistry at the tissue–shell interface in these minute, 80–100 μm sized larvae.

Owing to the lack of understanding of the processes leading to first larval shell formation, here we investigate the initial dynamics of calcium uptake and deposition using in vivo confocal and polarized microscopy in mussel (*Mytilus edulis*) larvae. We use calcein labeling to test whether calcium carbonate is accumulated intracellularly in vesicles and later deposited into the shell and how rapidly calcium is accumulated within the first larval shell. Further, we investigate the chemical composition of the extracellular CS using ion-selective microelectrodes for pH, $CO_3^{2-}$ and $Ca^{2+}$. Finally, we subject larvae to OA conditions and investigate how changes in seawater carbonate chemistry impact carbonate chemistry at the CS and correlate with shell growth and dissolution. We find no evidence for intracellular calcification via large intracellular vesicles (>1 μm), but demonstrate very rapid appearance of calcein label in the shell. Larvae are able to manipulate carbonate chemistry at the CS to enhance rates of calcium carbonate precipitation, but cannot maintain favorable conditions when they are exposed to OA. Decreases in pH, $[CO_3^{2-}]$ and $\Omega_{arag}$ at the site of calcification under OA conditions lead to reduced shell length growth and, ultimately, shell dissolution.

## Results

### No evidence for intracellular larval calcification.
To understand how the larval PD I shell is formed during the first 2 days of life

and whether intracellular accumulation of calcium carbonate in large vesicles plays a role in shell formation, we conducted several time series experiments at 17 °C. Calcium accumulation of whole larvae was measured at ten developmental time points using flame photometry (Fig. 1a). During the first 22 hours post fertilization (hpf, trochophore stage), we detected a moderate, ca. 3-fold increase in larval calcium content. In contrast, an exponential, 250-fold increase in $[Ca^{2+}]$ was observed between 22–40 hpf. This increase correlated with the first appearance of calcein labeled particles attached to the organic cover (the periostracum) of each valve of the growing shell field at 21 hpf (Fig. 1b, c) and progressive expansion of the shell valves, as indicated by extensive calcein labeling of newly formed shell material (Figs. 1 and 2). Once calcein positive structures were deposited on the organic shell cover (Fig. 1f, g), subsequent calcein label appeared in meandering bands that interconnected the initially deposited particles and extended the shell valves across the larval body (Figs. 1h and 2h–j). Using crossed polarized microscopy we could demonstrate that the growing shell is birefringent from 22 hpf onwards (Fig. 1i, j, Supplementary Movie 1), indicating the presence of a high fraction of crystalline aragonite. Presence of aragonite in larval *M. edulis* shells from 22 hpf (at 18 °C) onwards has previously been observed using XRD[21] and early veliger shells of other bivalve species also consist of aragonite[22, 23]. Video recordings of swimming and rotating larvae under crossed polarized light indicate that the entire shell is birefringent (Supplementary Movie 1). These findings indicate that if ACC is deposited onto the growing shell, the majority must rapidly (within minutes to hours) transform into crystalline aragonite.

When culturing larvae in calcein containing filtered seawater (FSW), numerous large (0.5–3 μm diameter) calcein positive vesicles were observed in all cells (Fig. 1b, c). In order to determine, whether these vesicles contribute calcium carbonate to the shell, we conducted a series of calcein pulse-chase experiments. When staining larvae with calcein during 0–20 hpf and growing them afterwards in calcein-free seawater until 48 hpf (Fig. 2a–d), none of the vesicular calcein label appeared in the shell, indicating that larvae do not accumulate a significant intracellular calcium reservoir prior to shell formation. This is in accordance with our larval calcium content measurements (Fig. 1a). When larvae were labeled for 15–180 min with calcein during the initial shell formation phase (21–30 hpf) and then cultured in non-labeled FSW for several more hours (Fig. 2b), the label rapidly (within <15 min, Fig. 2g, h) appeared in the shell and in many intracellular vesicles. However, none of this intracellular label was found in shell material formed after the calcein labeled FSW had been removed (Fig. 2e, f). This also suggests that, unlike in echinoderm larvae[15], large intracellular calcein positive vesicles do not serve as a calcium reservoir for shell formation. The most parsimonious explanation for our findings is that rapid transepithelial transport of substrates ($Ca^{2+}$, $HCO_3^-$) to the CS enables extracellular shell formation. Transport of calcification substrates through transepithelial pathways (transcellular and paracellular) has been observed in several biological systems[24–26]. The rapid appearance of calcein label in the growing shell (<15 min, Fig. 2g, h) suggests that mussel larvae epithelia are relatively "leaky", allowing the relatively large calcein molecule (0.6 kDa) to pass via paracellular pathways to the CS. Similarly, calcification substrates ($Ca^{2+}$, $HCO_3^-$/$CO_3^{2-}$) could take a similar route from seawater to the CS. This would make the calcification process vulnerable to fluctuations in extracellular carbonate chemistry at the CS.

### Carbonate chemistry at the CS is influenced by OA.
In order to characterize the carbonate chemistry at the CS, microelectrode measurements were performed in the extracellular space below

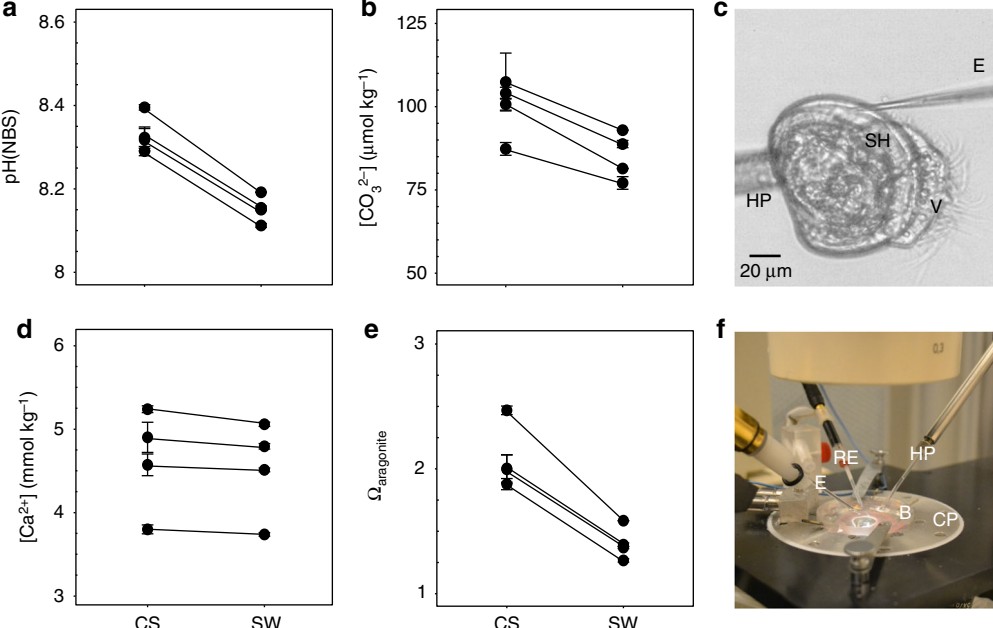

**Fig. 3** Carbonate chemistry parameters in the calcification space of PD I larvae from Exp 3. **a** $pH_{NBS}$. **b** $[CO_3^{2-}]$. **c** Image of larva attached to a holding pipette (HP) with the tip of the measuring electrode (E) visible at the site of calcification (CS), below the shell (SH), V = velum. **d** $[Ca^{2+}]$. **e** $\Omega_{arag}$. **f** Setup for microelectrode measurements showing inverted microscope, the electrode, holding pipette, water bath (B), reference electrode (RE) and cooling plate (CP). Plotted data are the mean ± s.d. from $N = 4$ fertilizations and 5 larvae for each treatment and fertilization

the growing shell edge in PD I veliger larvae (Fig. 3a–f). Our measurements demonstrate that early veliger larvae have active control over the carbonate chemistry of the fluid adjacent to the growing shell, which is in contrast to the situation in adult mussels, where extrapallial fluid pH and $\Omega_{arag}$ are lower than the respective seawater values[27]. Under control conditions, pH at the site of calcification ($pH_{CS}$) was found to be elevated by ca. 0.2 pH units with respect to the surrounding seawater pH ($pH_{SW}$, $v = 325$, $p < 0.01$, Fig. 3a), $[CO_3^{2-}]$ by 14 µmol kg$^{-1}$ ($t = -9.8338$, $p < 0.01$, Fig. 3b) and $[Ca^{2+}]$ by ca. 0.1 mmol kg$^{-1}$ ($v = 185$, $p < 0.05$, Fig. 3d). These conditions favor calcification and prevent dissolution, as they lead to a significant increase in $\Omega_{arag}$ at the tissue–shell interface: $\Omega_{arag}$ is increased from seawater values of ca. 1.4 to 2.2 at the site of calcification ($v = 325$, $p < 0.01$, Fig. 3e). Using Eq. (1), this increase in $\Omega_{arag}$ would enhance calcification rates by a factor of ca. 2.7. The mechanisms underlying this pH regulatory effort are yet unexplored in bivalves. Rapid aragonite deposition in mussel larvae has been correlated with increased activity of carbonic anhydrase, but it is unclear if it plays a role in CS carbonate chemistry regulation[21]. However, based on larval oyster gene expression data, all typical pH regulatory proteins that are commonly utilized to control extracellular and intracellular pH in marine metazoans are expressed in bivalve larvae[28].

When larvae were reared under simulated OA conditions, $pH_{CS}$ and $pH_{SW}$ declined linearly with increasing $pCO_2$ (Fig. 4a). The offset between $pH_{CS}$ and $pH_{SW}$ could be maintained up to $pCO_2$ values of about 1,500–2,000 µatm. In fact, $\Delta[H^+]$ between SW and CS remained constant between 500 and 1,500 µatm, indicating that acidification cannot be compensated by increased rates of proton removal from the CS. While $[Ca^{2+}]_{CS}$ remained constant over the entire range of $pCO_2$ levels tested, $[CO_3^{2-}]_{CS}$ strongly declined with $[CO_3^{2-}]_{SW}$, from ca. 100 to 39 µmol kg$^{-1}$ (Fig. 4d). The sigmoidal collapse of $[H^+]$ gradients indicates the sensitivity threshold of the extracellular $pH_{CS}$ regulation machinery (Fig. 4c). Together, these changes led to a strong exponential decline in $\Omega_{arag}$ at the CS, albeit with a significant positive offset between SW and CS: undersaturation ($\Omega_{arag} < 1$) at the CS occurred at a seawater $pCO_2$ ca.

999.5 µatm while undersaturation in SW was already reached at ca. 723.6 µatm (Fig. 4f). This finding illustrates the ability of mussel larvae to offset impacts of OA at the mineral interface to some degree, but also demonstrates the limited capacity of the regulatory system. Changes in carbonate chemistry at the CS likely impact both, catalytic function of extracellular pH sensitive enzymes involved in shell matrix processing, as well as rates of carbonate precipitation and dissolution[12, 29].

### Low $\Omega_{arag}$ at the CS leads to reduced growth and dissolution.

In accordance with the $CO_2$ induced progressive decrease in $\Omega_{arag}$ at the tissue–mineral interface, calcification rates (i.e. shell length at the PD I veliger stage) decreased in a linear fashion (Fig. 5a), a finding that is in accordance with several other studies on mussel and oyster larvae[3, 6–8]. However, mussel larvae were able to form fully developed PD I veliger shells in all treatments, even at seawater $pCO_2$ values that resulted in $\Omega_{arag}$ undersaturation at the CS. While a range of studies has demonstrated that mortality and rates of shell deformation are increased at very high seawater $pCO_2$ (>2,000 µatm), some larvae are clearly able to overcome the adverse carbonate chemistry conditions—potentially via modulation of organic matrix components that stabilize precipitated carbonate material or via production of chemically more resistant nanogranular building blocks[30]. In addition, it has been empirically demonstrated that bivalves continually experience dissolution of the internal surface of their shell, even under conditions where $\Omega_{arag} > 1$[20], which is due to the inherently higher solubility of biogenic carbonates. Therefore, elevation of $\Omega_{arag}$ at the CS in mussel larvae may be crucial to help offset shell dissolution. In order to investigate, whether $\Omega_{arag} < 1$ at the CS leads to shell dissolution, as recently shown for oyster larval stages[7], we conducted another calcein labeling experiment: larvae were reared for 21 h under control conditions, labeled with calcein for 3 h and then exposed to acidified conditions for 3 days (Fig. 5c) to determine, whether labeled shell material would dissolve once deposited. Shell fluorescence intensity of the labeled shell parts followed a sigmoidal pattern (Fig. 5b), with significant decreases

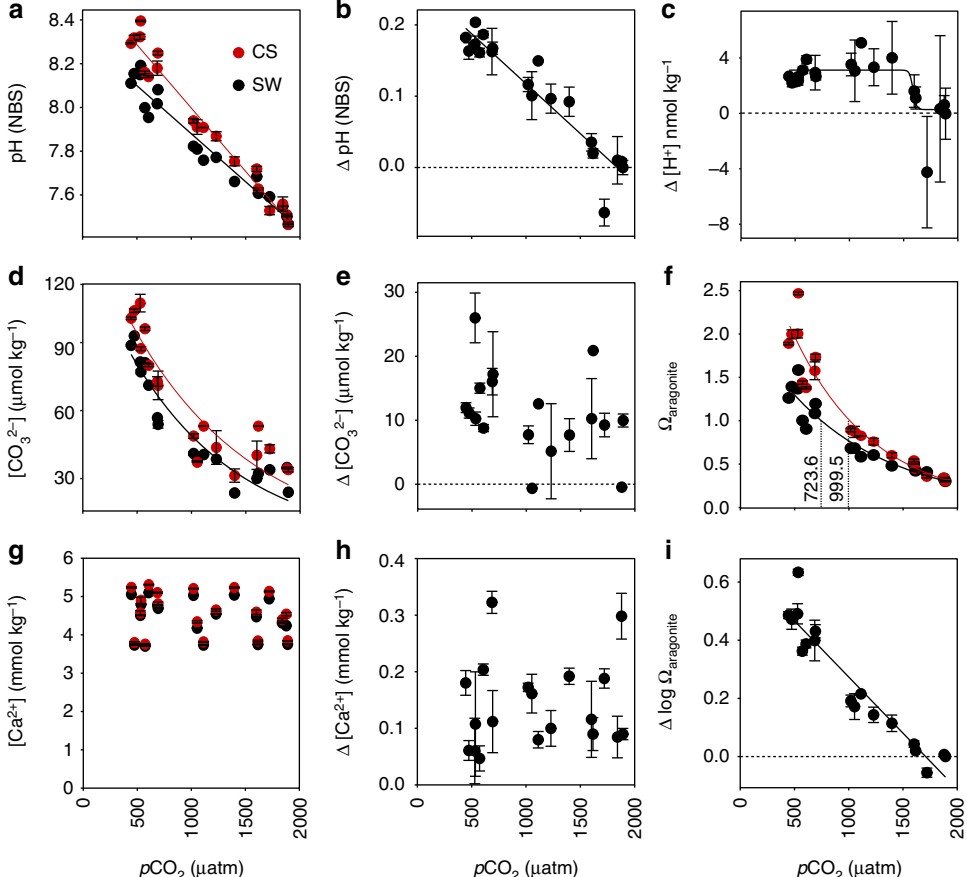

**Fig. 4** Impact of OA, in black on carbonate chemistry parameters at the site of calcification in red from Exp 3. **a** $pH_{NBS}$ **b** $\Delta pH_{NBS}$ (CS-SW) **c** $\Delta$ [H$^+$] **d** [CO$_3^{2-}$] **e** $\Delta$ [CO$_3^{2-}$] (CS-SW) **f** $\Omega_{arag}$ **g** [Ca$^{2+}$] **h** $\Delta$[Ca$^{2+}$] **i** $\Delta$ log $\Omega_{arag}$. Plotted data are the mean + s.d. from $N = 4$ fertilizations and 5 larvae for each treatment and fertilization.

in fluorescence (by ca. 20%) likely related to decreases in shell thickness[31] only detectable at very high levels of OA (2000 and 2500 µatm). This finding is in contrast to strong dissolution effects observed in larval oysters at lower $pCO_2$, but indicates the high degree of biological control over mineral integrity in mussels even under strongly acidified conditions[7].

Ion and acid–base regulation are metabolically costly, as is the production of organic matrix components for shell formation[32, 33]. It is unclear what the costs for maintenance of the observed gradient in $\Omega_{arag}$ are, but the inability of mussels to increase the proton gradient between SW and CS with increasing levels of OA indicates a fixed ion regulatory capacity at the tissue-mineral interface. Our results suggest that observed reductions in calcification with OA could be primarily attributed to decreased rates of CaCO$_3$ precipitation at decreasing $\Omega_{arag}$ at the CS. In summary, we provide novel evidence of how ocean acidification influences $\Omega_{arag}$ at the larval tissue-shell interface. Significant, but ultimately limited ion regulatory capacity to overcome acid–base challenges at the CS contributes to vulnerability of mussel larvae to ocean acidification. Although mussel larvae have been shown to successfully complete larval development in coastal habitats that are already CO$_2$ enriched today[34], future changes in coastal ocean carbonate chemistry may rapidly deplete this limited degree of ion regulatory control[10].

## Methods
**Animal collection and larval culture.** Adult *Mytilus edulis* were collected in April–June 2015 and 2016 from Kiel Fjord, outside GEOMAR. Spawning was induced by rapidly increasing water temperatures to 18–24 °C and gametes were

collected in filtered seawater (FSW, 0.2 µm). A total of >50 separate fertilizations were obtained during this period to conduct four experiments (Exp 1–4). 30 min following fertilization, embryos were checked for presence of polar bodies and embryo cleavage and placed in 2–10 L glass bottles filled with FSW at a density of 100 embryos per mL (Exp 1) or 10 embryos per mL (Exp 2–4) at a temperature of 17 °C. Culture bottles were gently and continuously aerated using pressurized air supplied through plastic tubing, connected to 10 mL plastic Pasteur pipettes. Following PD I veliger development, FSW was daily supplemented with *Rhodomonas salina* to reach a final concentration of 10,000 cells ml$^{-1}$. For the OA experiments (Exp 3, 4), FSW was continuously aerated with CO$_2$-enriched air (400, 800, 1,250, 1,600, or 2,000 µatm). Cultures were maintained at a natural salinity of 14–15‰.

**Carbonate chemistry measurements and calculations.** Carbonate chemistry samples were collected from the culture bottles just before adding embryos to the bottles. Samples were collected in 52 ml Duran Schott glass bottles with glass stoppers and preserved by the addition of 10 µl of saturated HgCl$_2$ solution. Dissolved inorganic carbon ($C_T$) was analyzed using an AIRICA $C_T$ analyzer (Marianda, Germany). Seawater carbonate system speciation ($pCO_2$, total alkalinity, [HCO$_3^-$], [CO$_3^{2-}$]) was calculated from $C_T$ and pH$_{NBS}$ using CO2SYS[35] with dissociation constants for KHSO$_4$[36] and the carbonate system (K1 and K2)[37, 38]. Certified seawater reference materials (batch 142; Scripps Institution of Oceanography, University of California, San Diego, CA, USA) were used to ensure accuracy of $C_T$ measurements. Saturation state of aragonite was calculated using [Ca$^{2+}$] and [CO$_3^{2-}$] from microelectrode measurements and the apparent solubility product for aragonite calculated for the respective salinity and temperature conditions according to Mucci[39].

**Calcium accumulation.** Exp 1: To investigate calcium accumulation, larvae between 4 and 40 hpf from $N = 5$ replicate families were collected on a 20 µm mesh and transferred to a 2 ml Eppendorf tube using a plastic Pasteur pipette. For each development stage, 90,000–300,000 individuals were collected. Samples were centrifuged and the supernatant was discarded. The samples were then vortexed in 500 µl distilled water and centrifuged for 10 min at 21,800 r.p.m. The supernatant was discarded and the larval pellet was dissolved in 500 µl 10 M HCl. Samples were

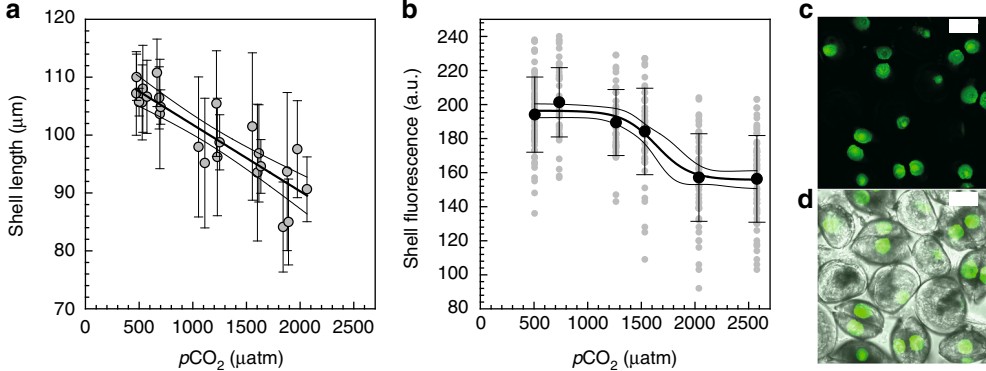

**Fig. 5** Impact of $CO_2$-driven seawater acidification on calcification in PD I larvae from Exp 3 and 4. **a** Shell length at PD I, veliger larval stage. Data are means ± s.d. from $N = 5$ fertilizations and 10–25 animals per treatment and fertilization. Linear regression: shell length (µm) = 113.3–0.0115 $pCO_2$ (µatm), $R^2 = 0.74$, $F_{(1,22)} = 63$, $p < 0.001$. **b** Calcein fluorescence intensity of larval shell portions formed during 21–24 hpf imaged following 3 day exposure to OA. Data are the mean ± s.d. from one experiment. Sigmoidal regression: fluorescence intensity (a.u.) = 155.8 + 40.6/(1 + exp (−($pCO_2$ − 1639)/−156.1)), $R^2 = 0.36$, $F_{(3,338)} = 64$, $p < 0.001$. **c** Calcein fluorescence confocal projections of larvae used to measure shell dissolution. **d** Merged fluorescence and transmission images. Scale bars: 75 µm

analyzed using a flame photometer (EFOX 5053, Eppendorf) that was calibrated using urine standards (Biorapid). Calcium concentrations were normalized per larvae following a background subtraction for the HCl larvae were dissolved in. Larvae from each developmental stage were sampled and fixed in 4% PFA buffered to pH 8.1. Larval samples were subsequently analyzed to assess mineralization using an inverted microscope (Leica DMi8) equipped with crossed polarized light filters.

**Calcein pulse-chase**. Exp 2: Larval cultures were stained using 50 mg l$^{-1}$ calcein label at pH 8.2 in FSW for various time periods, ranging from continuous culture (4–48 hpf) in calcein labeled FSW ($>N = 30$ separate experimental fertilizations) to 15–180 min calcein labeling intervals during key phases of the shell formation process ($N = 3$ separate experiments per time interval). Following labeling, animals were concentrated while constantly submerged in FSW using 20 µm nylon mesh sieves and gently washed using FSW until no traces of calcein were visible. Subsequently, animals were either directly imaged or cultured in unlabeled FSW and sampled a few hours later. For confocal microscopy, larvae were placed on microscopy slides suspended in FSW and covered with glass coverslips supported by ca.30-µm-thick fibers as spacers. Larvae were then imaged in vivo with a Zeiss LSM510 using a Plan Neofluar 40×/1.3 oil objective and a 505–530 nm band-pass filter at an excitation wavelength of 488 nm. Images (10–30 µm thick z-stacks of 1 µm optical sections) were collected within 20–30 min at 20 °C and processed to create 3D projections using Zeiss LSM510 software. Typically, larvae remained viable for >60 min under these conditions.

**Calcifying space carbonate chemistry**. Exp 3: Using ion-selective electrodes, H$^+$, $CO_3^{2-}$ and $Ca^{2+}$ concentration were measured below the surface of the shell in 48 h veliger larvae (5 animals from each treatment for each of $N = 4$ separate fertilizations, Supplementary Table 1). Microelectrode construction and measurements were performed as described previously[18, 40, 41]. Briefly, borosilicate glass capillary tubes (GB150F-8P, Science Products) were pulled using a DMZ-Universal puller (Zeitz Instruments) to a diameter of 1–3 µm and silanized with dimethyl chlorosilane (Sigma-Aldrich) at 200 °C for 4 h. The pH selective microelectrodes were back filled with a KCL based electrolyte (300 mM KCl, 50 mM NaPO$_4$, pH 7) and front loaded with a H$^+$ ionophore (H$^+$ ionophore III, Sigma Aldrich). The calcium selective microelectrodes were back filled with a KCl based electrolyte (200 mM KCl, 2 mM CaCl$_2$.2H$_2$O) and front loaded with a Ca$^{2+}$ ionophore (Ca$^{2+}$ ionophore II, Sigma Aldrich). The carbonate selective microelectrodes back filled with a CaCO$_3$ based electrolyte (19.1 g l$^{-1}$ Na$_2$B$_4$O$_7$*10H$_2$O adjusted with 100 mM HCl to pH 9.0, amended with 0.2 mM CaCO$_3$ after which the pH was again brought to 9.0) and front loaded with a $CO_3^{2-}$ ionophore (N,N,-dioctyl-3α,12α-bis(4-tri-fluoroacetylbenzoxy)-5β-cholan-24-amide, TFAP-CA, Sigma Aldrich). To measure pH, carbonate and calcium gradients, the larvae were transported to the microscope set up in Duran Schott glass bottles (500 ml). Measurements were performed within 2 h of transporting larvae and no changes in carbonate chemistry in the bottles were observed. For each $pCO_2$ treatment, measurements from 5 larvae per fertilization were obtained within 30 min. Larvae were concentrated submerged in respective FSW media using a 50 µm mesh and placed into a perfusion chamber mounted on an inverted microscope (Axiovert 135; Zeiss) at a density of 100 ml$^{-1}$. The microscope was equipped with a temperature controlled stage, connected to a water cooling system to maintain water bath temperatures at constant 17 °C±0.1 °C. Larvae were held in position using a holding pipette with a tip diameter of ca. 25 µm, to which a slight vacuum was applied (Fig. 3c). The ion-selective probe was

mounted on a remote-controlled micro-manipulator and was introduced beneath the shell from the side of the growing edge (Fig. 3f) and typically the larvae reacted by closing their valves. However, when open, the microelectrode was inserted 3–4 µm from the shell edge until the glass tip of the microelectrode reached the inorganic (hard) inner surface of the shell. The microelectrode was held in position in this space near the inner shell surface, between the shell and tissue. Measurements were conducted using a high impedance amplifier (World Precision Instruments, FD223). Stable mV readings were obtained within 5–10 s. The Nernstian property of each microelectrode (ca. 52–54 mV for pH electrodes and ca. 24–27 mV for [$CO_3^{2-}$] and [$Ca^{2+}$] electrodes) was measured by placing the microelectrode in a series of seawater solutions with a reference Ag/AgCl electrode (pH range 7.5–8.3, [$CO_3^{2-}$] range 20–130 µmol kg$^{-1}$, and [$Ca^{2+}$] range 0.1–10 mmol kg$^{-1}$). Calibration curves were prepared for each new set of electrodes using seawater of varying pH (using HCl and NaOH) for pH electrodes, $CO_2$ bubbled seawater with [$CO_3^{2-}$] ranging between 20–130 µmol kg$^{-1}$ (calculated using pH and $C_T$) for [$CO_3^{2-}$] microelectrodes and artificial seawater with [$Ca^{2+}$] ranging between 0.1 and 10 mmol kg$^{-1}$. New electrodes were built for each fertilization. With this setup, we were able to resolve a minimum difference in pH of ~0.002, [$CO_3^{2-}$] of ~1 µmol kg$^{-1}$ and [$Ca^{2+}$] of ~0.02 mmol kg$^{-1}$. A subsample of larvae from each treatment were sampled directly prior to microsensor measurements and fixed in 4% PFA buffered to pH 8.1 for determination of shell length at PD I. Thirty individuals for each treatment from $N = 5$ separate fertilizations were photographed using a stereomicroscope at ×100 magnification (Leica M165 FC) equipped with a Leica DFC 310 FX camera and shell length was measured using ImageJ.

**Shell dissolution**. Exp 4: A larval culture (one fertilization) was labeled with calcein from 21–24 hpf under control conditions (400 µatm). Subsequently, larvae were gently washed in FSW and distributed in culture vessels exposed to $pCO_2$ treatments of 400, 800, 1,250, 1,600, 2,000 and 2,500 µatm for 3 days. Samples were frozen at −20 °C for 2 weeks and analyzed using confocal microscopy as described above. No shell fluorescence loss occurred following the freezing procedure. To quantify loss of fluorescence from shell parts formed at 21–24 hpf in response to OA, 3D projections of one shell valve of 45–70 animals per $pCO_2$ level were generated and fluorescence intensity determined for a shell area of 100–150 µm$^2$.

**Data analyses**. Data were analysed using paired $t$-tests following tests for normality and homogeneity using Shapiro-Wilks test and Bartlett test respectively. If assumption for normality were not met, a non-parametric Wilcoxon signed-rank test was applied. To study the effect of OA on carbonate system speciation and [$Ca^{2+}$] of the CS vs. SW, shell length and dissolution, regression analysis was conducted where linear model functions for each parameter were compared by applying an ANOVA (Supplementary Tables 2 and 3). Statistical analyses were conducted using R (Version 3.3.2, R Development Core Team, R: http://www.R.org/. 2011).

**Data availability**. Data can be accessed through PANGAEA database (https://doi.pangaea.de/10.1594/PANGAEA.881869).

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

## Acknowledgements

This study was funded by the European Union's Seventh Framework Programme [FP7] ITN project "CACHE" under REA grant agreement #[605051]13, the Kiel Excellence Cluster "Future Ocean" and the GEOMAR seed-funding grant "microsensors 70090/06". We are grateful to Claas Hiebenthal and Ulrike Panknin for Kiel Marine Organism Culture Centre (KIMOCC) support.

## Author contributions

K.R. and F.M. designed the study. K.R., M.Y.H., J.T., M.B. and F.M. conducted experiments. K.R. and F.M. analysed the data and wrote the manuscript. All authors contributed to manuscript revisions.

## Additional information

**Competing interests:** The authors declare no competing financial interests.

