## [Peer Review File · Nature Communications]

Reviewers' comments:

Reviewer #1 (Remarks to the Author):

The well-written manuscript by Ramesh et al presents the first set of microelectrode measurements ($[Ca^{2+}]$, pH, and $[CO_3^{2-}]$) of the extrapallial fluid (EPF) of larval mussels under normal and CO₂-acidified conditions – which allows for the first calculations of the aragonite saturation state of the larval mussel EPF. For this reason, alone, I believe that the paper merits publication in a topical, high impact journal such as NCC. Furthermore, the authors show 2 very important things: (1) EPF pH is elevated relative to seawater in larval mussels (contrary to in some adult species, where it has been down to be less than seawater pH) and (2) the pH and thus saturation state of this fluid decreases with decreasing seawater pH – thus providing a mechanism by which OA effects larval mussel calcification.

That said, I do have some suggestions that I think would materially improve the manuscript:

(1) The title 'Mussel larvae modify calcifying fluid carbonate chemistry to counter ocean acidification' seems misleading. From the data presented, the larvae seem to be elevating EPF pH to calcify, not necessarily to counter OA. This pH elevation at the site of calcification may make them more resilient to the effects of OA, but to interpret the data as evidence that they are elevating EPF pH to specifically actually counter OA would require that they are making more of an effort to elevate EPF pH (and thus greater delta pH) under the higher pCO₂ conditions – which I do not see evidence of. In fact, I see the opposite – delta pH (and delta H⁺) decreases with increasing pCO₂ – suggesting they are putting less effort (not more) to elevate saturation state at the site of calcification under acidified conditions.

(2) The authors repeatedly state that adult bivalves are robust to OA stress, but I do not see the evidence for this. In fact, Ries et al (2009, *Geology*, attached) and Gazeau et al (cited in ms) and Waldbusser et al (cited in ms), and even some of the authors of the present manuscript, have all shown that most bivalves (with the exception of mussels) exhibit some of the most negative responses to OA of all of the major calcifying groups. In a comparative study looking at 5 species of adult bivalves, Ries et al (2009) showed that while oysters, hard clams, soft clams, and bay scallops all show a highly negative response to OA (400 – 2800 ppm), mussels showed no statistically significant response to this range of pCO₂. Perhaps a more accurate statement would be that adult mussels have been shown to be robust to OA stress – a statement that would also be more consistent with their findings (elevated EPF pH within mussels).

(3) The authors initiate a discussion about the effects of dissolution on biogenic carbonates. However, they cite only the rate equation ($r = k(\Omega_{arag} - 1)$) developed for abiogenic carbonates (based on a solubility product derived for abiogenic aragonite). This equation requires/dictates that biogenic carbonates dissolve at $\Omega < 1$ and precipitate at $\Omega > 1$. However, Ries et al's (2016, *GCA*, attached) recently published study on the dissolution kinetics of biogenic carbonates, which included 4 species of bivalves (mussels, oysters, hard clams, and soft clams) showed that, across temperatures of 10-25 C, dissolution of bivalves' calcite and aragonite actually begins at saturation states well above 1 – most likely due to the biogenically precipitated carbonates being fundamentally more soluble than the inorganic carbonates on which the solubility products in the Ω value are based (either due to entrapment of organic molecules, ACC or other metastable, more soluble phases of CaCO₃). Thus, we now know that the inorganic rate equation described above does not describe the dissolution kinetics of biogenic carbonates. Furthermore, because of the ' $k(\Omega - 1)$ ' term in that equation, the k -coefficient, that the authors define as a 'biocalcification constant', is incapable of adjusting the value of Ω that divides dissolution from calcification. Since Ries et al (2016) has empirically derived the equation governing the dissolution kinetics of both calcitic mussel shells and aragonitic clam shells (spanning mineralogy of both the aragonitic larval and calcitic adult stages of mussels), perhaps the authors should consider discussing this work in the context of their discussion of mussel shell

dissolution. This new understanding of bivalve and, specifically, mussel shell dissolution kinetics (i.e., the empirical dissolution curves for various bivalves and the observation that dissolution is occurring even in oversaturated conditions) should also inform the interpretation of their results for the following reasons: (1) in >600 ppm pCO₂ waters, the authors' data include specific values for EPF saturation state < 1, and we now have empirical data relating these low saturation states to dissolution rates of calcitic and aragonitic bivalve shell and (2) even in low pCO₂ waters (i.e., oversaturated), we now know that mussels are constantly fighting dissolution of their exterior shell because biogenic carbonates are fundamentally more soluble than abiogenic versions of the same carbonate mineral. This emphasizes the importance of elevating EPF pH (now documented for larval mussels in the present paper) in order to produce new shell beneath the mantle, which allows them to offset the ever-occurring dissolution of their shell exterior.

(4) Since the authors accomplished the impressive feat of directly measuring the Ca and CO₃⁼ ions in the EPF to directly calculate (for the first time that I am aware) the actual saturation state of the EPF, I would strongly suggest that they plot EPF in Figure 3 on a non-logarithmic scale to show what the actual saturation state of the EPF is. It is presently plotted only on a log-scale – thus, it is hard for readers to decipher the absolute saturation state.

J Ries
5 May 2017

Reviewer #2 (Remarks to the Author):

The manuscript entitled 'Mussel larvae modify calcifying fluid carbonate chemistry to counter ocean acidification' examines the mechanisms of mussel larvae shell formation. The article examines how larvae can continue to form shells under reduced carbonate availability induced by ocean acidification. The study uses an approach of measuring pH and carbonate beneath the growing shell using ion-sensitive microelectrodes. These techniques have been applied before to adult mussels to show that the pH within the extrapallial fluid below the shells is controlled by biochemical mechanisms to maintain shell formation internally irrespective of the outside seawater pH and carbonate undersaturation. However additionally the study combines the use of microelectrodes with calcein staining of intracellular vesicles to identify the uptake of calcium into the early forming larval shell. The novelty here is the first use of these combined techniques applied to early larvae shell formation. These mechanisms are poorly understood and is vital to predict vulnerabilities of mollusc shell formation under ocean acidification induced low carbonate environments. The major claim of the article is the ability of the mussel larvae to control the pH and carbonate concentrations above the seawater levels in the calcification space, I consider this to be well justified by the evidence presented. Additionally of interest, which appears evidenced by the polarised light microscopy images, is the claim by the authors that the calcium is not taken from the intracellular vesicle stores as such in sea urchin larvae. The figure indicating the calcium accumulation from the calcein positive intracellular vesicles is not clear to follow. I consider Supplementary Figure S1 of the calcein pulse chase experiments, in vivo confocal microscopy to show this more clearly. I have suggested to the authors to use these images for Figure 1 as an alternative for easier digestion by the reader.

This study is of great importance to scientists in the field as the mechanisms of early larval shell formation are not clearly understood. This approach would be of interest to scientists in the field of biomineralization and ocean acidification as a tool to identify vulnerable early life stages of molluscs. The data is technically sound, the authors have been very thorough in their experimental approach according to my understanding of the methods and figure legends using N=4 families and 5 larvae for each determination of microelectrode measurements and N=5 families 10-25 individual larvae for calcification in PD1 larvae. I have minor suggestions for further clarification of replication in Figure legend 1 for the calcium accumulation and in vivo larval shell formation data. I

consider this article provides strong evidence for the major claims of the ability of the mussel larvae to control the pH and carbonate concentrations above the seawater levels in the calcification space. This is highlighted by the evidence nicely presented in Figures 2 and 3. The article does argue for rapid crystalline aragonite formation within the calcification space without a ACC precursor phase, due to the birefringent nature of the forming larval shell from 22 hours post fertilisation. I do question the evidence for these conclusions and consider that the authors should further clarify their reasoning for these claims. The study has not explicitly tested for ACC 0-22 hours post fertilisation within the shell only referring to their imaging using the polarised light microscopy. With this in mind, I consider this article to have the general criteria for publication in Nature Communications in respect to the novelty in approach, technical rigour and strong evidence of the ability of the larvae to modify calcifying fluid carbonate chemistry. However, I would suggest to the editor that further clarification of the theories of phase transition of ACC to rapid crystalline aragonite early shell formation is required before accepting for publication. I consider this discussion to take away from the nicely presented story of the title of the article 'Mussel larvae modify calcifying fluid carbonate chemistry to counter ocean acidification' and could be omitted. Reducing the emphasis of the paper to suggest 'rapid crystalline aragonite early shell formation'. I leave this to the editor.

To the authors please see questions to specific points below:

Line 75 – The authors comments that 'if no ACC is deposited the majority must be rapidly transformed into crystalline aragonite.' Did the authors test for phase of crystal or ACC during the early larvae formation time points (0-22 hpf)? For example applying techniques such as SEM-EBSD? This is an exciting finding if indeed the authors are correct that the birefringent nature of the forming larval shell appears to transform rapidly to crystalline aragonite without an ACC precursor phase.

Lines 88-90 – how can the authors be sure that the transepithelial transport of Ca^{2+} and HCO_3^- occurs to the calcification space for rapid extracellular shell formation? Could these ions not be taken from seawater or formed using proteins in the extrapallial fluid? Please clarify.

Lines 246 – the statistical testing appears to be appropriate for data analysis.

Figure 1 legend – to be consistent how many families were used for calcium accumulation for in vivo larval shell formation, the authors have suggested N=4 means and SD are these individual larvae? For Figure 2 and 3 the authors suggest N=4 fertilisations and the results are for 5 individual larvae per fertilisation.

Figure 1 – the choice of images for figure 1 are difficult to follow for the evidence of calcium accumulation from the calcein positive intracellular vesicles. I consider Supplementary Figure S1 of the calcein pulse chase experiments, in vivo confocal microscopy to show this more clearly. I would suggest to the authors to use these images for Figure 1 as an alternative for easier digestion by the reader.

Reviewer #3 (Remarks to the Author):

Ramesh et al. provide an interesting and unique study of calcification in mussel larvae and how it responds to changes in seawater carbonate chemistry. Overall this is a nice study that is well written and should be publishable in Nature Comms. The application of these techniques to bivalve larvae is pretty cool, and provides some much needed data in understanding aspects of early bivalve larval calcification.

My biggest concerns with the MS in its current state is the lack of some key literature, the incomplete development of some arguments, and some possible misinterpretation of other literature and results here. I believe all these should be easily resolvable, and if so, I see no reason to prevent this manuscript from being published here. I have highlighted these concerns below. The largest issue, in my opinion, to deal with is the conclusion of supporting no change in

energy consumption under OA, and I have highlighted those issues in more detail below. The authors should look very carefully at the cited paper they note, as there are some serious methodological flaws and lack of acknowledgement of the larvae in that study essentially being in extremely poor condition. I had reviewed an earlier version of the MS, and my comments along with those of two other reviewers were ignored by the editor and the paper published without most of the issues addressed.

But back to the paper here, I hope the authors are able to address the issues I outlined below, and I look forward to seeing this manuscript in print in the near future.

Paragraph starting on line 27, see Mount et al. 2004 for evidence of intracellular calcification in adult oysters. I have often wondered if this was simply an repair mechanism versus a primary calcification route, but this citation should be included and addressed in this context.

Line 66- would you say the "organic cover" is the periostracum?

Line 88- Are the authors suggesting that direct exchange between seawater and the CS is not possible here? There are a number of older papers that have looked at some of the energetics associated with moving substrates transepithelial and the limitations in bivalves (in adult stages that are considerably slower than the larval calcification here). If I am following the argument here, the labeled Ca would need to cross the epithelial once first to become part of the haemolymph, then again into the calcification space. Maybe that is not what the authors intended to write, but this should be clarified. A series of papers have addressed the Ca transport issue previously, and it would be worthwhile to acknowledge this earlier work. Jordey 1953, Suzuki and Nagasawa 2001, Carre et al. 2006. There is not yet consensus in the literature, but there are compelling arguments for unrealistic costs associated with transepithelial transport of calcification substrates. Previous work on shell isotopes in bivalve larvae also suggest that a far greater proportion of shell C is derived from seawater at this stage versus respiratory carbon (Waldbusser et al. 2013).

Line 95- Is it also possible that the EPF in adults is lower because more respiratory carbon is shunted into the shell versus remaining in the EPF? I would not disagree with the base argument that the data shows some enhancement of pH and saturation state, but it seems odd that a far less developed larvae would have this ability to regulate the calcification space, then it is lost as an adult.

Line 102- There has been some work showing carbonic anhydrase activity in bivalves, I believe? This would provide the mechanism for elevating chemistry at the CS.

Line 134- This is an important conclusion and could use some additional clarification in the sense that dissolution in itself can typically only decrease the thickness of shells, not shell lengths. Shells in bivalve larvae are extended via additions to the periostracum, then the shell is mineralized from the inside surface.

Line 142- Be careful with citing this paper, if one looks closely at the data, it is clear that the larvae in those experiments, across the board, were highly energetically strained, and thus one would be unable to differentiate energetic effects (all the larvae looked as if it was entirely starved). Secondly, the labeled Ca experiments in that study did not use adequate corrections or controls for the equilibrium exchanges of radiolabeled Ca between water and shell.

While I agree with the general findings of the author's study here, the interpretation that somehow things are simply smaller without due to some energetic cost, is tenuous. As noted above, shell extension is different than calcification (while the two processes work together to make shell), the first, or as one may interpret, a measure of size would in fact likely be directly tied to the energetics. The second, calcification, or shell thickness, would be more impacted by the saturation

state in the CS. This was demonstrated by Gaylord et al. 2011 in mussel larvae, shell thickness and shell length did not vary in lock step under OA. While it appears that there is little ability to control CS conditions with changes to SW chemistry, the data in panel e figure 3 may indicate that there is a continued and maybe (unknown) cost to maintain. My point is that without other measurements (respiration rates, biochemical composition, etc), it is not certain that there is no energetic cost to maintain the offset as conditions become less favorable for precipitation. I believe the authors can provide a more balanced interpretation with appropriate caveats to address this.

Reviewer 1 comments:

(1) ‘The title ‘Mussel larvae modify calcifying fluid carbonate chemistry to counter ocean acidification’ seems misleading. From the data presented, the larvae seem to be elevating EPF pH to calcify, not necessarily to counter OA. This pH elevation at the site of calcification may make them more resilient to the effects of OA, but to interpret the data as evidence that they are elevating EPF pH to specifically actually counter OA would require that they are making more of an effort to elevate EPF pH (and thus greater delta pH) under the higher pCO₂ conditions – which I do not see evidence of. In fact, I see the opposite – delta pH (and delta H⁺) decreases with increasing pCO₂ – suggesting they are putting less effort (not more) to elevate saturate state at the site of calcification under acidified conditions.’

RESPONSE: Good point. We suggest an alternative manuscript title:

‘Mussel larvae modify calcifying fluid carbonate chemistry to promote calcification.’

(2) ‘The authors repeatedly state that adult bivalves are robust to OA stress, but I do not see the evidence for this. In fact, Ries et al (2009, Geology, attached) and Gazeau et al (cited in ms) and Waldbusser et al (cited in ms), and even some of the authors of the present manuscript, have all shown that most bivalves (with the exception of mussels) exhibit some of the most negative responses to OA of all of the major calcifying groups. In a comparative study looking at 5 species of adult bivalves, Ries et al (2009) showed that while oysters, hard clams, soft clams, and bay scallops all show a highly negative response to OA (400 – 2800 ppm), mussels showed no statistically significant response to this range of pCO₂. Perhaps a more accurate statement would be that adult mussels have been shown to be robust to OA stress – a statement that would also be more consistent with their findings (elevated EPF pH within mussels)’

RESPONSE: Good point again, we rephrased this section to indicate that in comparison to adults, larval stages in mussels are the most vulnerable to environmental / OA stress. We also added a sentence to indicate that bivalves in general are one of the more sensitive taxa to ongoing OA.

Line 5-6: ‘Especially in rapidly calcifying larval stages, mechanisms of shell formation are largely unexplored – yet these are the most vulnerable life stages’

Line 17-20: ‘Adult life stages of several bivalve species have been shown to be relatively vulnerable to OA. However, bivalve larvae are particularly sensitive and respond negatively through [...]’

(3) ‘The authors initiate a discussion about the effects of dissolution on biogenic carbonates. However, they cite only the rate equation ($r = k(\Omega_{\text{arag}} - 1)$) developed for abiogenic carbonates (based on a solubility product derived for abiogenic aragonite). This equation requires/dictates that biogenic carbonates dissolve at omega less than 1 and precipitate at omega greater than 1. However, Ries et al’s (2016, GCA, attached) recently published study on the dissolution kinetics of biogenic carbonates, which included 4 species of bivalves (mussels, oysters, hard clams, and soft clams) showed that, across temperatures of 10-25 C, dissolution of bivalves’ calcite and aragonite actually begins at saturation states well above 1 – most likely due to the biogenically precipitated carbonates being fundamentally more soluble than the inorganic carbonates on which the solubility products in the omega value are based (either due to entrapment of organic molecules, ACC or other metastable, more soluble phases of CaCO₃). Thus, we now know that the inorganic rate equation described above does not describe the dissolution kinetics of biogenic carbonates. Furthermore, because of the ‘ $k*(\omega - 1)$ ’ term in that equation, the k-coefficient, that the authors define as a ‘biocalcification constant’,

is incapable of adjusting the value of omega that divides dissolution from calcification. Since Ries et al (2016) has empirically derived the equation governing the dissolution kinetics of both calcitic mussel shells and aragonitic clam shells (spanning mineralogy of both the aragonitic larval and calcitic adult stages of mussels), perhaps the authors should consider discussing this work in the context of their discussion of mussel shell dissolution. This new understanding of bivalve and, specifically, mussel shell dissolution kinetics (i.e., the empirical dissolution curves for various bivalves and the observation that dissolution is occurring even in oversaturated conditions) should also inform the interpretation of their results for the following reasons: (1) in >600 ppm pCO₂ waters, the authors' data include specific values for EPF saturation state < 1, and we now have empirical data relating these low saturation states to dissolution rates of calcitic and aragonitic bivalve shell and (2) even in low pCO₂ waters (i.e., oversaturated), we now know that mussels are constantly fighting dissolution of their exterior shell because biogenic carbonates are fundamentally more soluble than abiogenic versions of the same carbonate mineral. This emphasizes the importance of elevating EPF pH (now documented for larval mussels in the present paper) in order to produce new shell beneath the mantle, which allows them to offset the ever-occurring dissolution of their shell exterior.'

RESPONSE: Thanks for pointing our attention towards this quite interesting new paper. We have decided to leave eq. 1 in the manuscript (with a note that it relates to precipitation of inorganic carbonates) to enable a consistent comparison and discussion with other key papers on bivalve larval responses to ocean acidification (Waldbusser et al 2013, Waldbusser et al 2015). However, we have incorporated findings from Ries et al. 2016 in the discussion section:

Line 51-52: 'The chemical composition of this space should govern calcification and dissolution rates (r) for inorganic carbonates according to...'

Line 55-56: 'However, shell dissolution for biogenic aragonite in bivalves has been empirically observed to occur under conditions where $\Omega_{\text{arag}} > 1$, likely due to the higher solubility of biogenic carbonates'

Line 149-152 'In addition, it has been empirically demonstrated that bivalves continually experience dissolution at the surface of their shell, even under conditions where $\Omega_{\text{arag}} > 1$ ¹⁹, which is due to the inherently higher solubility of biogenic carbonates. Therefore, elevation of Ω_{arag} at the CS in mussel larvae may be an important mechanism to offset the continual dissolution of their internal shell surface.'

(4) 'Since the authors accomplished the impressive feat of directly measuring the Ca and CO₃⁼ ions in the EPF to directly calculate (for the first time that I am aware) the actual saturation state of the EPF, I would strongly suggest that they plot EPF in Figure 3 on a non-logarithmic scale to show what the actual saturation state of the EPF is. It is presently plotted only on a log-scale – thus, it is hard for readers to decipher the absolute saturation state.'

RESPONSE: We have amended the figure (now Figure 4) to include the absolute values of saturation state as suggested.

Reviewer 2 comments:

- (1) 'Line 75 – The authors comments that 'if no ACC is deposited the majority must be rapidly transformed into crystalline aragonite.' Did the authors test for phase of crystal or ACC during the early larvae formation time points (0-22 hpf)? For example applying techniques such as SEM-EBSD? This is an exciting finding if indeed the authors are correct that the birefringent nature of the forming larval shell appears to transform rapidly to crystalline aragonite without an ACC precursor phase.'

RESPONSE: Within our study, we can establish that if ACC is a precursor to shell formation, it must rapidly crystallize based on the birefringence patterns observed in this study. Polarized microscopy reveals that the entire shell material is birefringent from 21hpf onwards suggesting that the bulk of shell carbonates are crystalline (see also supplementary video). This is also in accordance with a range of other studies that found aragonite in larval bivalve shells (Lee et al 2006, Kudo et al 2010, Yokoo et al 2011, Thompson et al 2014). Rapid crystallization is important, as the shell serves as attachment point for musculature of the feeding and swimming apparatus (Waldbusser et al 2015).

In addition, we have performed a detailed analysis of shell carbonates in larval *M. edulis* and other bivalve larvae using *in vivo* confocal Raman microscopy (for the first time ever). Our results suggest that there is no detectable amount of ACC in mussel larval shell carbonates between 21 – 48 hpf (the onset of calcification until the D-veliger stage). We exclusively sampled Raman spectra characteristic for crystalline aragonite. (Kirti Ramesh, Frank Melzner, Gernot Nehrke et al *in prep*). The paper will be submitted very soon and we can make an advanced manuscript draft available if requested. Of course this does not exclude the potential existence of ACC precursor phases that crystallize very rapidly.

- (2) ‘Lines 88-90 – how can the authors be sure that the transepithelial transport of Ca^{2+} and HCO_3^- occurs to the calcification space for rapid extracellular shell formation? Could these ions not be taken from seawater or formed using proteins in the extrapallial fluid? Please clarify.’

RESPONSE: To clarify, transepithelial transport which consists of both transcellular (mediated by primary and secondary ion transport proteins, as well as ion selective channels) and paracellular pathways, which has been demonstrated to be responsible for transfer of biomineralization substrates (Ca^{2+} and HCO_3^-) in many animal taxa (Wheeler 1992, Bleher & Machado 2004, Stumpp et al 2012, Tambutté et al 2012). So far, there is no evidence for direct seawater routes to the CS. In addition, there is currently no evidence for protein induced calcification within the EPF in bivalves with studies suggesting little calcium transport through calcium binding proteins (Silanpää et al 2016). Alternatively, larvae may employ endocytosis of seawater to transport substrates to the CS as demonstrated in echinoderms. However, the calcein labelling experiment within this study suggests that mussel larvae do not utilize a vesicular pathway for calcification, lending little support to this hypothesis. Therefore, the rapid movement of calcification substrates to the CS to support larval calcification rates likely occurs via transepithelial pathways. Our calcein pulse chase experiments also revealed that calcein (molecular mass: 0.6 kDa) rapidly (within minutes) flooded the extracellular space of larvae, appearing in nascent shell material already after 15 minutes. This indicates that mussel larval epithelia are quite ‘leaky’ for large molecules, similar to recent findings in echinoderm larvae (Stumpp, Hu, Melzner et al. 2012 PNAS, see Fig.2). It is thus likely that calcification substrates also can quickly enter the mussel larvae from the seawater via such leaky paracellular pathways. We added a sentence:

Lines 104-109: ‘Transport of calcification substrates through trans-epithelial pathways (trans-cellular and para-cellular) has been observed in several biological systems^{24, 25, 26}. The rapid appearance of calcein label in the growing shell (<15 minutes, Fig. 2c,d) suggests that mussel larvae epithelia are relatively ‘leaky’, allowing the relatively large calcein molecule (0.6 kDa) to pass via para-cellular pathways to the CS. Similarly, calcification substrates (Ca^{2+} , $\text{HCO}_3^-/\text{CO}_3^{2-}$) could take a similar route from seawater to the CS.’

- (3) Figure 1 legend – to be consistent how many families were used for calcium accumulation for *in vivo* larval shell formation, the authors have suggested N=4 means and SD are these individual larvae? For Figure 2 and 3 the authors suggest N=4 fertilisations and the results are for 5 individual larvae per fertilisation.’

RESPONSE: We clarified this by including details regarding the replicates for Figure 1 in the legend. The legends for Figures 2-3 have been maintained as they were interpreted correctly by the reviewers.

‘Figure 1 | Calcium accumulation and in vivo larval shell formation (Exp 1 and 2) (a) larval calcium accumulation during the first 48hpf, N=4 fertilisations, means and SD.’

- (4) ‘Figure 1 – the choice of images for figure 1 are difficult to follow for the evidence of calcium accumulation from the calcein positive intracellular vesicles. I consider Supplementary Figure S1 of the calcein pulse chase experiments, in vivo confocal microscopy to show this more clearly. I would suggest to the authors to use these images for Figure 1 as an alternative for easier digestion by the reader.’

RESPONSE: We have incorporated this suggestion into the revised manuscript by including supplementary Fig S1 in the main manuscript as the new Figure 2.

Reviewer 3 comments:

- (1) ‘Paragraph starting on line 27, see Mount et al. 2004 for evidence of intracellular calcification in adult oysters. I have often wondered if this was simply an repair mechanism versus a primary calcification route, but this citation should be included and addressed in this context.’

RESPONSE: This reference has been included in the manuscript in the suggested context.

Line 38-40: ‘Presence of crystalline calcite and deposition onto the shell has been observed in oyster hemocytes during shell repair^{12, 13}.’

- (2) Line 66- would you say the “organic cover” is the periostracum?

RESPONSE: The organic cover is the periostracum as described by Kniprath (1981) and we have edited the manuscript to reflect this.

Line 78-81: ‘In contrast, an exponential, 250-fold increase in [Ca²⁺] was observed between 22-40hpf. This increase correlated with the first appearance of calcein labeled particles attached to an organic cover (the periostracum) of each valve of the growing shell field at 21hpf (Fig. 1b,d) and progressive expansion of the shell valves, as indicated by extensive calcein labeling of newly formed shell material (Fig. 1c, S1).’

- (3) ‘Line 88- Are the authors suggesting that direct exchange between seawater and the CS is not possible here? There are a number of older papers that have looked at some of the energetics associated with moving substrates transepithelial and the limitations in bivalves (in adult stages that are considerably slower than the larval calcification here). If I am following the argument here, the labeled Ca would need to cross the epithelial once first to become part of the haemolymph, then again into the calcification space. Maybe that is not what the authors intended to write, but this should be clarified. A series of papers have addressed the Ca transport issue previously, and it would be worthwhile to acknowledge this earlier work. Jordey 1953, Suzuki and Nagasawa 2001, Carre et al. 2006. There is not yet consensus in the literature, but there are compelling arguments for unrealistic costs associated with transepithelial transport of calcification substrates. Pervious work on shell isotopes in bivalve larvae also suggest that a far greater proportion of shell C is derived from seawater at this stage versus respiratory carbon (Waldbusser et al. 2013).’

RESPONSE: We believe our response to comment (2) of reviewer 2 addresses this comment. In addition, Jodrey (1953) and Carre et al (2006) in fact lend support for transepithelial transport of calcium for shell formation, where Jodrey (1953) observe rapid turnover of a small concentrations of ⁴⁵Ca in the mantle during shell deposition and Carre et al (2006) propose the role of ion transport proteins such as Ca²⁺-ATPases in calcification.

- (4) 'Line 95- Is it also possible that the EPF in adults is lower because more respiratory carbon is shunted into the shell versus remaining in the EPF? I would not disagree with the base argument that the data shows some enhancement of pH and saturation state, but it seems odd that a far less developed larvae would have this ability to regulate the calcification space, then it is lost as an adult.'

RESPONSE: The reviewer raises an interesting point. The traditional view point is that marine metazoans need to have higher CO₂ partial pressures in their extracellular fluids (such as EPF and hemolymph) in order to excrete metabolic CO₂ via diffusion (see e.g. Melzner et al. 2009 Biogeosciences, Heinemann et al. 2012 G-cubed). This is apparently different in mussel larvae, and also in larval echinoderms (see comment 2 to reviewer 2, discussion in Stumpp et al. 2012 PNAS), indicating that these life stages can operate with extremely shallow CO₂ diffusion gradients from extracellular fluid to seawater, which might be beneficial for calcification. The additionally enhanced capacity of mussel larvae to regulate carbonate chemistry at the CS might have evolved to cope with the very high calcification rates typical for these life stages (e.g. Waldbusser et al. 2013, Thomsen et al. 2015 Biogeosciences) .

- (5) 'Line 102- There has been some work showing carbonic anhydrase activity in bivalves, I believe? This would provide the mechanism for elevating chemistry at the CS.'

RESPONSE: Good suggestion, we added a sentence discussion work by Medakovic (2000) on mussel larvae carbonic anhydrase activity during initial shell formation.

Line 122-124: 'Rapid aragonite deposition in mussel larvae has been correlated with increased activity of carbonic anhydrase, but it is unclear if it plays a role in CS carbonate chemistry regulation²⁰.'

- (6) 'Line 134- This is an important conclusion and could use some additional clarification in the sense that dissolution in itself can typically only decrease the thickness of shells, not shell lengths. Shells in bivalve larvae are extended via additions to the periostracum, then the shell is mineralized from the inside surface.'

RESPONSE: Good suggestion, we have rephrased this section and better explained our conclusions:

Line 156-158: 'Shell fluorescence intensity of the labeled shell parts followed a sigmoidal pattern (Fig. 5b), with significant decreases in fluorescence (by ca. 20%) likely related to decreases in shell thickness³¹ only detectable at very high levels of OA (2,000 and 2,500 \$\mu\$ atm).'

- (7) 'Line 142- Be careful with citing this paper, if one looks closely at the data, it is clear that the larvae in those experiments, across the board, were highly energetically strained, and thus one would be unable to differentiate energetic effects (all the larvae looked as if it was entirely starved). Secondly, the labeled Ca experiments in that study did not use adequate corrections or controls for the equilibrium exchanges of radiolabeled Ca between water and shell.'

RESPONSE: We have taken into consideration the comments of the reviewer regarding the Frieder et al (2017) paper and decided to not speculate with respect to energetic costs of calcification (see below).

'While I agree with the general findings of the author's study here, the interpretation that somehow things are simply smaller without due to some energetic cost, is tenuous. As noted above, shell extension is different than calcification (while the two processes work together to make shell), the first, or as one may interpret, a measure of size would in fact likely be directly tied to the energetics. The second, calcification, or shell thickness, would be more impacted by the saturation state in the CS. This was demonstrated by Gaylord et al. 2011 in mussel larvae, shell thickness and shell length did not vary in lock step under OA. While it appears that there is little ability to control CS conditions with changes to SW chemistry, the data in panel e figure 3 may indicate that there is a continued and maybe (unknown) cost to maintain. My point is that without other measurements (respiration rates, biochemical composition, etc), it is not

certain that there is no energetic cost to maintain the offset as conditions become less favorable for precipitation. I believe the authors can provide a more balanced interpretation with appropriate caveats to address this.'

RESPONSE: We removed our initial suggestion of potentially unaltered energetic costs of shell formation. We included the Gaylord et al. (2011) paper (line 156-158).

REVIEWERS' COMMENTS:

Reviewer #2 (Remarks to the Author):

I consider that the majority of points raised in the previous round of reviews have been satisfactorily addressed by the authors. Previously I raised concerns that the study had not explicitly tested for ACC 0-22 hours post fertilisation within the shell only referring to their imaging using the polarised light microscopy. The authors respond with clarification that there is no evidence for ACC, but rapid aragonite crystallisation occurs evidenced by the birefringent patterns observed post 21 hpf, however admit 'Of course this does not exclude the potential existence of ACC precursor phases that crystallize very rapidly'. Following this clarification by the authors, a careful re-read of the manuscript and taking into account the video S1, I believe that the manuscript satisfactorily addresses this original concern. I look forward to reading the authors follow- on publication on in vivo confocal Raman microscopy of larval *M. edulis*. Additionally I raised concerns around the lack of clarification of how the transepithelial transport of Ca^{2+} and HCO_3^- occurs to the calcification space for rapid extracellular shell formation. The later was also a concern of another reviewer and I consider this has now been adequately addressed with the citation of several papers discussing how the larvae epithelia are relatively 'leaky' allowing transport of large molecules via para cellular pathways. The now more suitably titled 'Mussel larvae modify calcifying fluid carbonate chemistry to promote calcification' manuscript claims to be the first study to use ion-sensitive micro-electrodes at the site of calcification (CS) in combination with calcein staining in the bivalve larvae to investigate how changes in the seawater carbonate chemistry alter those carbonate chemistry parameters in the CS. This study is highly novel and of great interest to the field of biomineralisation. The manuscript provides convincing evidence for the rapid calcification of aragonite seemingly constrained to the CS, rather than intracellularly. Additionally the novel use of ion-sensitive microelectrodes provides compelling evidence of the biomineralisation mechanisms of the first larval aragonite shell formation, indicating higher level controls by the larvae to increase pH and carbonate to maintain much higher levels of aragonite saturation state. This is evidenced to promote shell growth rather than the expected shell dissolution under changing levels of ocean acidification. I consider this manuscript to be influential in the field of bivalve biomineralisation and ocean acidification, particularly in the future studies of underlying mechanisms of early larval shell formation. In light of the revisions of the manuscript I would therefore suggest to the editor that the manuscript is well suited for publication in Nature Communications.

Reviewer #3 (Remarks to the Author):

The author's did a fairly nice job in revising the manuscript in response to both reviews. I feel they struck a nice balance and while there are still a few points I would disagree with, I don't believe it is worth holding up the publication (nor do I believe the issues are fully resolved here or in the greater literature). Therefore, those points are sure to stimulate further dialogue and discussion helping to move the science ultimately forward (I hope).